# Evaluation of a commercial pressure cooker for the preparation of agar media for a diagnostic microbiology laboratory

Joseph E. Rubin[ID][1]*, Florence Huby[1], Roshan P. Madalagama[ID][1,2], Shyamali de Alwis[2], Melinda Wyshynski[1], Rasika Jinadasa[3]

1 Department of Veterinary Microbiology, University of Saskatchewan, Saskatoon, Canada, 2 Division of Bacteriology, Veterinary Research Institute, Peradeniya, Sri Lanka, 3 Department of Veterinary Pathobiology, University of Peradeniya, Peradeniya, Sri Lanka

* joe.rubin@usask.ca

## Abstract

The ability to prepare sterilized media is a critical capability of any microbiology lab. Diagnostic labs in low-resource settings, which lack autoclave facilities, are therefore severely limited in their ability to perform basic assays such as bacterial culture or biochemical tests. This investigation aimed to validate the use of a commercially available pressure cooker as an autoclave substitute to produce agar plates. First, a *Geobacillus stearothermophilus* biological indicator was used to confirm adequate sterilization. Next, the colony morphology of several important bacterial species were compared on MacConkey and 5% sheep's blood agar plates prepared using the pressure cooker with those made in an autoclave. Finally, disc diffusion susceptibility testing was performed to determine whether the sterilization method impacts the inhibitory zone diameters. Overall, the morphology of colonies was similar on media prepared in both ways; key phenotypic characteristics (lactose fermentation, colour, shape, hemolysis and smell) were the same. Kirby-Bauer disc diffusion test results were nearly identical. These findings indicate that a commercially available pressure cooker may be suitable to prepare media in low-resource laboratories.

## Introduction

The autoclave is a critical piece of equipment for any microbiology laboratory. The ability to prepare sterile media (agar plates, broths and biochemical tests) is essential for the growth and characterization of microbes in diagnostic, research and teaching contexts. An autoclave is a steam sterilizer, which kills microbes using high temperature and pressure by coagulating and denaturing proteins and damaging DNA [1,2]. While these harsh conditions are intended to kill microbes, they can also impact organic media through the formation of Maillard reaction products and the destruction of sugars [3]. Consequently, when preparing media, the length of exposure to heat

**Data availability statement:** All relevant data are within the manuscript. Most of the data is visual (photos displaying similarities in colony morphology between plates).

**Funding:** This study was funded in part by the University of Saskatchewan's Global Community Service Fund. There was no additional external funding received for this study.

**Competing interests:** The authors have declared that no competing interests exist.

and pressure must be calibrated to ensure that media are not undesirably altered in the process. Other methods of sterilization are often unpractical, ineffective or not suitable for preparing media. Filter sterilization is not amenable to large volumes, as the filters are expensive. Boiling is inexpensive, but ineffective for the destruction of some spores. Gamma irradiation requires specialized facilities and chemical sterilants (formaldehyde, ethylene oxide, hydrogen gas plasma and peracetic acid) are not suitable for media preparation [4,5]. Steam sterilization (autoclaving) is therefore uniquely suited to the in-house production of microbiological media. While previous studies have evaluated the use of a pressure cooker for sterilization in some contexts (ex. surgical and laboratory instruments, decontamination of biohazardous waste) the applicability of a pressure cooker for the preparation of agars used in a clinical diagnostic laboratory has not been evaluated [5–8]. A 2018 study evaluating the sterilizing ability of multiple brands of commercially available pressure cooker, found that only the Instant Pot was able to inactivate the *Geobacillus stearothermophilus* biological indicator, a critical quality control metric for laboratory autoclaves [6]. By evaluating the performance of agar produced in an Instant Pot, the current study represents the next logical step for implementing this method in a real-world setting.

In Sri Lanka, veterinary diagnostic services are provided by a network of 25 Veterinary Investigation Centres (VICs) nationwide, and by the Veterinary Research Institute (VRI) in Peradeniya (national reference laboratory for veterinary diagnostics). The VICs are responsible for disease investigations (farm visits and outbreak identification), diagnostic testing including post-mortem examination, bacteriology and serological testing and referral of samples to the central facility for advanced analysis. While some VICs have good physical infrastructure where well-trained diagnosticians provide high-quality service, others lack basic equipment and supplies, including functional autoclaves. Since 2023, Rubin and Priyantha have been working on a laboratory capacity development project to improve the quality of diagnostic services in Sri Lanka. The two main objectives are to produce custom training materials and manuals for human resources development as open-access resources (procedures manual volume 1: https://hdl.handle.net/10388/15571, procedures manual volume 2: https://hdl.handle.net/10388/16657), and to identify specific challenges faced by diagnosticians while finding creative and practical solutions. In March 2024, a broken autoclave shut down microbiological testing for months at one centre. In rural Sri Lanka, laboratories face financial and logistical barriers to the maintenance of operational autoclaves. This equipment is expensive (costing thousands to tens of thousands of Canadian dollars depending on the model), and funds to replace non-functional equipment are unavailable. Furthermore, a lack of service technicians in rural areas means that VIC staff must personally transport equipment to larger centres on the bus for repairs to be made. The objective of the current study is therefore to address this challenge by evaluating the ability of a commercially available pressure cooker to sterilize media without adversely impacting colony morphology. This validation study aimed to determine if the Instant Pot® could be used as an alternative sterilization method to prepare agar plates for use in a veterinary diagnostic laboratory when a steam autoclave is inaccessible.

## Materials and methods

In December 2024, an Instant Pot® (Ultra 60) was purchased ($CAD 169.96 + tax) for validation experiments. The user manual states that the device reaches a pressure of 10.2–11.6 psi [9], which is lower than a conventional autoclave where sterilization cycles reach 15 psi, a longer sterilization time is therefore required when using this device [2,10]. The first step was to develop a standard set of running conditions for sterilization based on a previously published protocol for use with this brand of pressure cooker [6]. Briefly, 500 ml of deionized water was placed in the pot to generate steam, and the "sterilize" function was used under a high-pressure setting. A commercially prepared *Geobacillus stearothermophilus* biological indicator kit (3M™ Attest™ Biological Indicator for Steam 1262, 3M Canada, London, Ontario) was placed in the Instant Pot® and cycle lengths of 45, 90, 120 and 150 minutes were trialed. The spore indicator was then incubated at and inspected for a colour change as per the manufacturer's instructions. If the *G. stearothermophilus* indicator showed that the endospores in the tube failed to germinate and grow, a second sterilization cycle under the same conditions was run to confirm killing.

To assess the impact of prolonged sterilization times on the performance of standard agar media used in diagnostic microbiology, TSA with 5% sheep's blood, MacConkey and Mueller-Hinton agars (Becton, Dickinson and Company, Sparks, Maryland) were prepared. Two flasks of each media were prepared in parallel; one was sterilized in a conventional autoclave at 121°C at 15 psi for 15 minutes (AC) and the other was pressure-cooked at high pressure for 150 minutes (IP). For all media, 120 ml volumes were prepared in 250 ml Erlenmeyer flasks. For blood and MacConkey agars, divided (quartered) 100 mm petri dishes were used so that bacterial colony morphology on conventionally autoclaved and pressure-cooked agars could be seen side by side. For Mueller-Hinton agar, 100 mm petri dishes were poured to a depth of 4 mm, the standard thickness for performing Kirby-Bauer disc diffusion testing [11].

Blood agar and MacConkey were inoculated with fresh overnight cultures of *Staphylococcus* ATCC 29213, *Enterococcus faecalis* ATCC 29212, *Escherichia coli* ATCC 25922 and *Pseudomonas aeruginosa* ATCC 27853. These cultures were selected to represent key Gram-positive and Gram-negative taxa with phenotypic diversity (lactose fermentation, hemolysis, colony colour and presence of a characteristic smell) commonly identified in the veterinary diagnostic laboratory. Plates were incubated overnight, and colony morphology was inspected to identify any differences between media sterilized using each method. The *S. aureus, E. coli,* and *P. aeruginosa* strains are also standard quality control organisms used in Kirby-Bauer disc diffusion testing, making them ideal for assessing the performance of Mueller-Hinton agar for susceptibility testing [12]. A panel of antimicrobials (amikacin, ampicillin, trimethoprim, ciprofloxacin, tetracycline and polymyxin B) were selected to represent 6 distinct antimicrobial classes; each organism was tested against drugs for which reference standards are published by the CLSI. Antimicrobial susceptibility testing was conducted in accordance with the CLSI standards, and the same McFarland 0.5 bacterial suspension was used to inoculate both the conventionally prepared and cooked media to ensure the same density of inoculum was used [11].

Finally, a second Instant Pot® of the same model was purchased and transported to the Veterinary Research Institute in Peradeniya, Sri Lanka, so that an on-the-ground feasibility trial could be conducted in March, 2025. Using the same sterilization cycle, 5% sheep's blood and MacConkey agars were prepared. Media were streaked with fresh overnight cultures of *E. coli*, *Salmonella* spp., and *Staphylococcus aureus* to confirm the performance of the agar.

## Results

Using a sterilization time of 150 minutes, the *G. stearothermophilus* biological indicator was killed on 2/2 attempts. Shorter-duration cooking failed to reliably sterilize the standard as growth was indicated by a change in the colour of the indicator to yellow.

Bacterial growth on both the conventionally sterilized and cooked MacConkey was as expected. Neither *S. aureus* nor *E. faecalis* grew. *E. coli* grew as pink, lactose-fermenting colonies on both preparations (Fig 1). *P. aeruginosa* grew as colourless colonies on both preparations, while the colonies were larger on the IP media compared to AC (Fig 1).

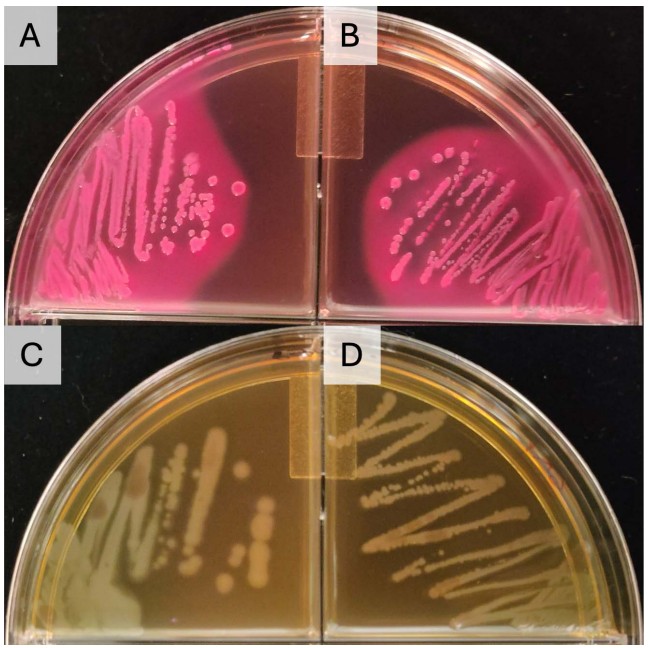

**Fig 1. Comparison of *E. coli* and *P. aeruginosa* colony morphology on MacConkey agar produced using a pressure cooker, and a conventional autoclave.**

Overnight cultures of *E. coli* on MacConkey agar sterilized in the IP (A) and a CA (B). Overnight cultures of *P. aeruginosa* on MacConkey agar sterilized in the IP (C) and CA (D).

On blood agar, only minor differences in colony morphology were observed (Fig 2). The growth of *E. coli* was indistinguishable on both media. *Pseudomonas aeruginosa* colonies were markedly larger on the IP compared to the AC. Other morphological characteristics of *P. aeruginosa* colonies were unaffected (colour, metallic sheen, lack of hemolysis, grape/floral aroma). Similarly, *E. faecalis* colonies were slightly larger on the IP as compared to the AC media. Minor differences were observed for *S. aureus*; the colonies were slightly larger on the IP media compared to those on the AC media. Additionally, the characteristic double-zone of hemolysis was enhanced on the IP media.

Overnight cultures of all four organisms are presented on blood agar. (1) *E. coli* a, b: pressure cooker (IP) conventional autoclave (CA) respectively, c, d – IP and CA backlit to see hemolysis. (2) *P. aeruginosa* a, b: IP and CA respectively, c, d: IP and CA tilted to see metallic sheen. (3) *E. faecalis* a, b: IP and CA respectively, c, d: IP and CA tilted to highlight colony colour. (4) *S. aureus* a, b: IP and CA respectively, c, d: IP and CA backlit to see hemolysis, e, f: CA and IP backlit to highlight hemolysis.

The zone diameters observed for all Kirby-Bauer disc diffusion testing fell within the reference intervals published by the CLSI (Table 1). For *P. aeruginosa,* the zone diameters were identical between media; although less green pigment was noted on the IP compared to the AC plates (Fig 3). For *S. aureus* and *E. coli*, the diameters of 2/5 and 3/6 discs were identical. The remaining results between media were very similar, differing by no more than 2 mm for *S. aureus* and 3 mm for *E. coli*.

(1) *P. aeruginosa* a, b: IP and CA respectively including amikacin, ciprofloxacin and polymyxin B discs. (2) *E. coli* a, b: IP and CA with ampicillin, amikacin and trimethoprim discs, c, d: IP and CA with ciprofloxacin, polymyxin B and tetracycline. (3) *S. aureus* a, b: IP and CA with ciprofloxacin and tetracycline, c, d: IP and CA with amikacin, ampicillin and trimethoprim.

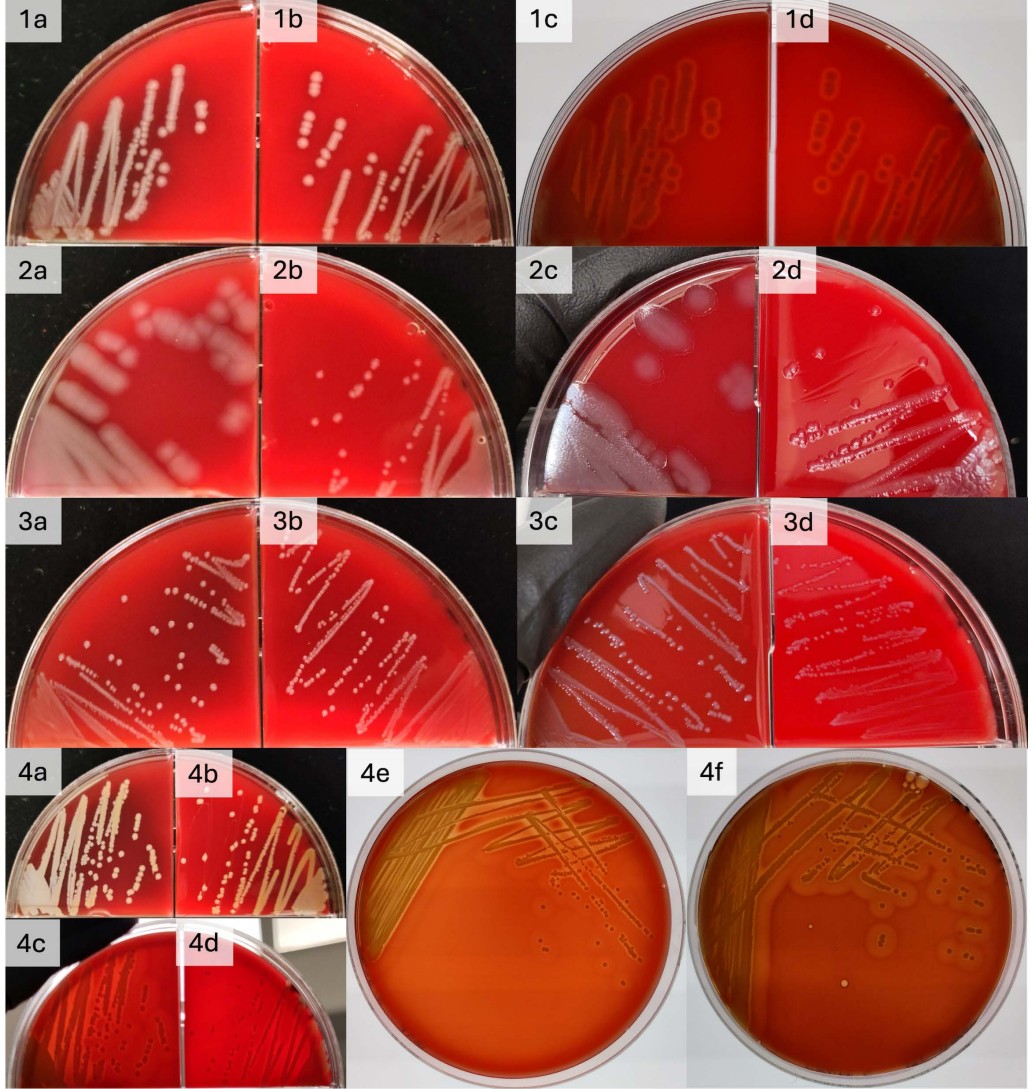

**Fig 2. Comparison of *E. coli, P. aeruginosa*, *E. faecalis* and *S. aureus* colony morphology on 5% sheep blood agar produced using a pressure cooker, and a conventional autoclave.**

Antimicrobial inhibitory zone diameters (in millimeters) for *S. aureus, E. coli* and *P. aeruginosa* for each of the 6 drugs tested on media prepared using either the standard sterilization method (autoclave) or the pressure cooker. The CLSI reference ranges for each drug/organism combination are included in parentheses in the autoclave column. Differences in the diameters observed between media are enumerated in the right-hand column of each organism. Shaded cells represent drug/organism combinations where resistance is intrinsic, and no references standard is available.

The results of the Instant Pot® field testing at the VRI campus in Sri Lanka were promising. The device was user-friendly, and the procedure was easy to follow. The *Salmonella*, *E. coli,* and *S. aureus* cultured grew as expected on the blood and MacConkey agar plates produced.

**Table 1. Comparison of Kirby-Bauer antimicrobial zone diameters using conventionally prepared, and pressure cooker sterilized media.**

| Antimicrobial | S. aureus ATCC 29213 | | | E. coli ATCC 25922 | | | P. aeruginosa ATCC 27853 | | |
|---|---|---|---|---|---|---|---|---|---|
| | Autoclave (ref range) | Pressure Cooker | Difference | Autoclave (ref range) | Pressure Cooker | Difference | Autoclave (ref range) | Pressure Cooker | Difference |
| Amikacin | 23 (20 –26) | 22 | 1 | 23 (19 –26) | 24 | 1 | 23 (20 –26) | 23 | 0 |
| Ampicillin | 35 (27-35) | 33 | 2 | 18 (15 –22 ) | 18 | 0 | | | |
| Trimethoprim | 26 (19 –26) | 26 | 0 | 26 (21 –28) | 26 | 0 | | | |
| Ciprofloxacin | 30 (22 –30) | 30 | 0 | 37 (29-38) | 40 | 3 | 30 (25-33) | 30 | 0 |
| Tetracycline | 27 (24 –30) | 26 | 1 | 23 (18 –25) | 24 | 1 | | | |
| Polymyxin B | | | | 17 (13 –19 ) | 17 | o | 17 (14 –18 ) | 17 | 0 |

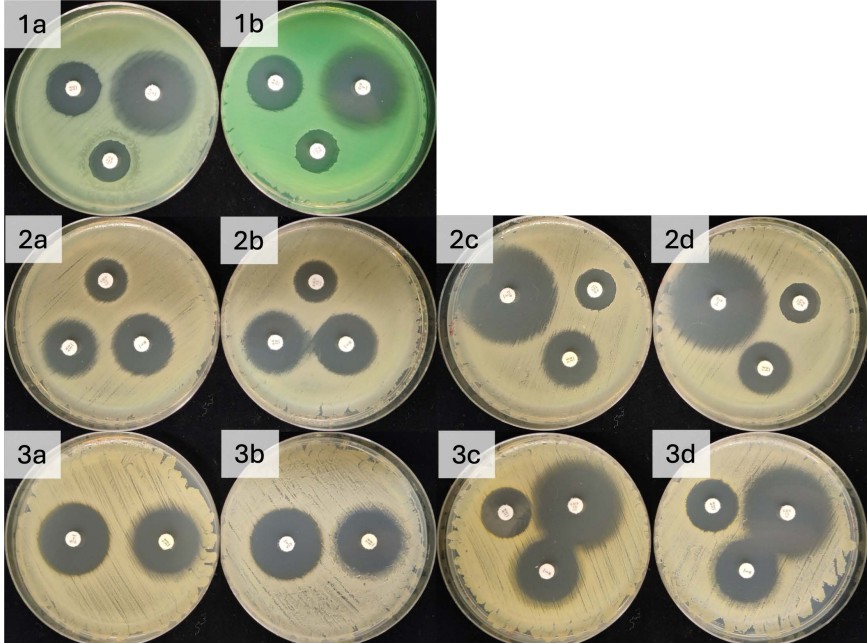

**Fig 3. Comparison of susceptibility test results of *E. coli*, *P. aeruginosa* and *S. aureus* on Mueller-Hinton agar produced using a pressure cooker, and a conventional autoclave.**

## Discussion

The results of this investigation are very encouraging. Consistent with a previously published study, a prolonged (150-minute) sterilization time is required to ensure killing of bacterial spores [6]. The blood and MacConkey agars, which were produced using the Instant Pot˚, performed very well compared to conventionally autoclaved agar. While there were small differences in colony morphology, most notably larger colonies on pressure-cooked vs. autoclaved media, the defining characteristics of each organism were unchanged, and we do not believe these differences would make bacterial identification more difficult. Indeed, in the case of *S. aureus,* the enhanced double zone of hemolysis seen on the pressure-cooked media could facilitate identification compared to conventional media.

When a veterinarian submits a sample for microbiological analysis, one of the key pieces of information they seek is a therapeutic recommendation based on susceptibility testing. It was therefore essential that Mueller-Hinton agar prepared using the Instant Pot˚ performed satisfactorily. The susceptibility test results obtained were very encouraging;

indeed, antimicrobial zone diameters measured on pressure-cooked media were consistent with the CLSI quality control ranges for the organisms tested and were very similar or identical to those observed on conventionally produced plates. The largest discrepancies were observed for *E. coli* with ciprofloxacin discs, and *S. aureus* with ampicillin discs; interestingly, these were also the two largest zone diameters observed, 40 and 33 mm, respectively. The non-linear relationship between drug concentration and distance from the antimicrobial disc means that even these discrepancies fall within the 95% confidence intervals for test performance used to establish reference ranges [13]. Furthermore, the CLSI guidelines state that there should be no more than a 3 mm discrepancy between repeated tests of the same organism when validating a new lot of media [14].

While the results of the current study are positive, the use of this device in a diagnostic setting does have an important limitation: volume. In the current study, up to 2 flasks of agar (120 ml agar, in a 250 ml Eylenmeyer flask or bottle) were prepared at a time. Limiting the volume of agar to no more than 50% of the volume of the container is essential to prevent boil over. Unfortunately, this small volume also greatly restricts the number of agar plates that can be produced simultaneously. However, diagnosticians in some VICs reported that they may only culture 5–10 samples/week, so this limitation may be less impactful in low volume, remote laboratory contexts. Furthermore, the centres with the lowest culture volumes tend to be in areas with the fewest laboratory resources and where the use of a pressure cooker could address a critical equipment deficiency.

This project also uncovered some logistical and product challenges that were not initially anticipated. First, it was impossible to have the Instant Pot® shipped to Sri Lanka directly or from a 3rd party vendor or to purchase this model locally. Therefore, it was necessary for the investigator to personally bring this appliance on the plane as checked luggage. Although the investigators discussed the possibility of purchasing a different brand of home pressure cooker in Sri Lanka, the validity of the study necessitated using the same model that was trialed in the Saskatoon (Rubin) laboratory. Furthermore, the study by Swenson et al. (2018) found that this brand was the only one which inactivate the *G. stearothermophilus* spores [6]. Finally, because of cost and procurement challenges, the biological indicator is not currently available at the VRI in Sri Lanka, so it would not have been possible to validate the use of another device.

Another initially unanticipated challenge is that because this model of Instant Pot® is designed for the North American market, it is rated to run on 110 V electricity versus the 230 V standard in Sri Lanka. This necessitated not only a plug adapter to convert a type B (North America and Japan) plug to fit a type G (Sri Lanka and United Kingdom) outlet but also a transformer to lower the voltage. This equipment was fortunately available locally, but would constitute and additional cost if it had to be purchased.

## Conclusions

Overall, it was found that the pressure cooker evaluated demonstrates promise to play the role of an autoclave, filling a vital equipment gap in rural Sri Lankan veterinary diagnostic laboratories. While this study did attempt to capture a range of clinically relevant bacterial species, it is possible that other fastidious organisms may grow differently. Future studies should assess the impacts of prolonged pressure cooking of media on the growth characteristics of other organisms of importance to veterinary medicine. Furthermore, in routine diagnostic testing, it would not be responsible to suggest using this method to prepare media other than those evaluated here without first validating it. It is essential to confirm that selective/differential properties of media are maintained and that key nutrients supporting the growth of fastidious organisms are not destroyed by the prolonged cooking period. Finally, due to the logistical challenges of using appliances designed for the North American market in other contexts, future studies should evaluate locally available appliances as autoclave alternatives.

## Author contributions

**Conceptualization:** Joseph E. Rubin, Roshan P. Madalagama, Rasika Jinadasa.

**Formal analysis:** Joseph E. Rubin.

**Funding acquisition:** Joseph E. Rubin.

**Investigation:** Joseph E. Rubin, Florence Huby, Roshan P. Madalagama, Shyamali de Alwis.

**Methodology:** Joseph E. Rubin, Melinda Wyshynski.

**Project administration:** Joseph E. Rubin, Florence Huby.

**Resources:** Rasika Jinadasa.

**Writing – original draft:** Joseph E. Rubin.

**Writing – review & editing:** Florence Huby.

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
