## [Decision Letter · Decision Letter 0]

18 Aug 2025

Dear Dr.Joseph E. Rubin

We look forward to receiving your revised manuscript.

Kind regards,

Tsegaye Alemayehu, Msc

Academic Editor

PLOS ONE

Journal Requirements:

“This study was funded in part by the University of Saskatchewan's Global Community Service Fund”

“This study was funded in part by the University of Saskatchewan's Global Community Service Fund”

4. We noted in your submission details that a portion of your manuscript may have been presented or published elsewhere. [An abstract describing this work was submitted for a conference presentation (poster) at the 2025 Canadian Society of Microbiologists annual conference. The abstract is not published and is not made available outside of the organizing committee.] Please clarify whether this [conference proceeding or publication] was peer-reviewed and formally published. If this work was previously peer-reviewed and published, in the cover letter please provide the reason that this work does not constitute dual publication and should be included in the current manuscript.

” The authors thank the University of Saskatchewan Global Community Service Fund for supporting this collaboration.”

“This study was funded in part by the University of Saskatchewan's Global Community Service Fund”

6. We note that your Data Availability Statement is currently as follows: [All data associated with this work is presented in the manuscript. Most of the data is visual (photos displaying similarities in colony morphology between plates).]

Reviewers' comments:

Reviewer's Responses to Questions

**Comments to the Author**

1. Is the manuscript technically sound, and do the data support the conclusions?

Reviewer #1: Yes

Reviewer #2: Yes

Reviewer #3: Yes

2. Has the statistical analysis been performed appropriately and rigorously?

Reviewer #1: Yes

Reviewer #2: No

Reviewer #3: I Don't Know

3. Have the authors made all data underlying the findings in their manuscript fully available?

Reviewer #1: Yes

Reviewer #2: Yes

Reviewer #3: Yes

4. Is the manuscript presented in an intelligible fashion and written in standard English?

Reviewer #1: Yes

Reviewer #2: Yes

Reviewer #3: Yes

Reviewer #1: Greetings

Dear Authors

1- In this manuscript, you wrote 6 times (We)! The rule of manuscript

writing is to avoid using (We). So you should delete (We) and use formal

scientific words (This study or The current study or The present study).

2- In this manuscript, you wrote 5 times (Our)! The rule of manuscript

writing is to avoid using (Our). So you should delete (Our) and use formal

scientific words (This study or The current study or The present study).

3- In introduction: you wrote two link of your books, kindly delete them because this considered advertising.

4- In methods, you did not write the date or period of current study for

example from May 2024 to December 2024.This is important scientifically.

Kind regards

Reviewer #2: This is a well-written and clearly structured manuscript that addresses a practical and relevant problem in microbiology laboratories in low-resource settings: the lack of functioning autoclaves for agar media preparation. The study is methodologically sound, with clear descriptions of procedures and an effective comparison between a commercial pressure cooker and a standard autoclave. The inclusion of field testing in a Sri Lankan veterinary diagnostic laboratory, along with Kirby-Bauer susceptibility testing, adds real-world value to the work.

However, the novelty of the concept is modest, as pressure cooker sterilization has been reported previously. The main contribution here lies in the practical validation within a specific veterinary diagnostic context and the demonstration of feasibility in a resource-limited field setting. To strengthen the manuscript, I recommend the following:

- Clearly position the work as a validation and implementation study, emphasizing what is new in your approach.

- Justify the limited range of bacterial species and media tested, and note that results should not be generalized to other organisms or media without further evaluation.

- Provide quantitative measurements for observed morphological differences (e.g., colony diameters) and include basic statistical comparisons for zone diameters.

- Consider adding a simple cost comparison between an autoclave and the Instant Pot® to highlight economic implications.

- Expand the limitations section to note the single model tested, potential nutrient degradation with prolonged heating, small production volumes, and electrical compatibility issues in different regions.

Reviewer #3: This article presents a highly practical and beneficial approach, particularly for laboratories that face challenges due to the lack of access to an autoclave. The alternative method proposed could serve as a valuable solution for resource-limited settings. However, further research is recommended, as highlighted by the authors, particularly regarding the use of different brands of pressure cookers in media preparation. Brands that are readily available and suitable for local contexts should be evaluated. In addition, the impact of electrical current capacity on the effectiveness and consistency of results should be examined, as it may influence performance. Lastly, the manuscript formatting should be reviewed to ensure it aligns with the journal’s submission guidelines.

**Do you want your identity to be public for this peer review?** For information about this choice, including consent withdrawal, please see our Privacy Policy

Reviewer #1: No

Reviewer #2: **Yes: ** Saqr Abushattal

Reviewer #3: No

---

## [Author Response · Author response to Decision Letter 1]

30 Sep 2025

Reviewer #1:

1- In this manuscript, you wrote 6 times (We)! The rule of manuscript writing is to avoid using (We). So you should delete (We) and use formal scientific words (This study or The current study or The present study).

• Thank you for identifying this overly casual style, "we" has been removed throughout.

2- In this manuscript, you wrote 5 times (Our)! The rule of manuscript writing is to avoid using (Our). So you should delete (Our) and use formal scientific words (This study or The current study or The present study).

• Thank you for identifying this overly casual style, "our" has been removed throughout.

3- In introduction: you wrote two link of your books, kindly delete them because this considered advertising.

• Respectfully we disagree with this point. We believe that referencing these manuals provides important context to our project overall. Furthermore, as both documents are 100% freely available open resources which are deposited into our University's library repository, there is no commercial product to advertise.

4- In methods, you did not write the date or period of current study for example from May 2024 to December 2024.This is important scientifically.

• We have added a second date to the manuscript. We started this investigation in December 2024 as stated, and then the final work in Sri Lanka was conducted in March, 2025.

Reviewer #2:

This is a well-written and clearly structured manuscript that addresses a practical and relevant problem in microbiology laboratories in low-resource settings: the lack of functioning autoclaves for agar media preparation. The study is methodologically sound, with clear descriptions of procedures and an effective comparison between a commercial pressure cooker and a standard autoclave. The inclusion of field testing in a Sri Lankan veterinary diagnostic laboratory, along with Kirby-Bauer susceptibility testing, adds real-world value to the work.

However, the novelty of the concept is modest, as pressure cooker sterilization has been reported previously. The main contribution here lies in the practical validation within a specific veterinary diagnostic context and the demonstration of feasibility in a resource-limited field setting. To strengthen the manuscript, I recommend the following:

- Clearly position the work as a validation and implementation study, emphasizing what is new in your approach.

• Thank you for this comment, . We have added additional context to highlight the novelty of using a pressure cooker to prepare media for a diagnostic laboratory (see lines 62-76). We have also added reference to this being a validation study (see line 103)

- Justify the limited range of bacterial species and media tested, and note that results should not be generalized to other organisms or media without further evaluation.

• Regarding the limited range of bacterial species tested, we agree that this is certainly a limitation. However these organisms do represent the most commonly encountered organisms by veterinary diagnostic laboratories, and the strains which we chose to test were those with zone diameter reference ranges published by the CLSI. We do have some justification on lines 121-126 as to why these species were chosen for our investigation.

• Regarding the applicability of our findings, this is indeed a very important point. We were careful to point out where our results are not generalizable in terms of both other organisms and media on lines 284-292.

- Provide quantitative measurements for observed morphological differences (e.g., colony diameters) and include basic statistical comparisons for zone diameters.

• Thank you for this comment. Both of suggestions are things which had been considered before submitting our initial draft.

o With respect to colony size, it is grossly apparent on figures 1 and 2 that there is variation in colony size even one the same media; those colonies growing in the most distant parts of the streak tend to be bigger than those growing close to the densest growth - this is a normal and expected phenotype for a bacterial culture. Consequently, the outcomes of statistical comparisons of colony measurements between media would be highly dependent on which colonies we chose to measure. For this reason we instead chose to use sectioned petri-dishes to directly compare conventionally prepared and pressure cooked media on the same plate; this allows us to see colonies which were grown under exactly the same conditions and for the same length of time side by side without the potentially distorting effects of photography (taking the picture from exactly the same distance, same lighting, holding the plate at the same angle to ensure that lighting is uniform etc.). We believe that the qualitative perspective the reader gains from viewing these images provides a more accurate representation of the differences/similarities observed than would a statistical comparison.

o With respect to zone diameters, the Clinical and Laboratory Standards Institute (CLSI) guidelines for the ATCC strain reference ranges which we used are intended to be used as a binary comparison; a control organism is classified as either within or outside the reference interval. Furthermore, it is recognized that a degree of variability between tests of the same organism (no more than 3mm) is expected when evaluating a new lot of media. We have included the reference ranges in table 1, the most important data from this experiment was that our results with both the pressure cooker and autoclaved media fell within the reference ranges and would therefore be considered up to the standards of a diagnostic microbiology laboratory.

- Consider adding a simple cost comparison between an autoclave and the Instant Pot® to highlight economic implications.

• We have added a notation on lines 81-82 providing a general sense as to the costs of a laboratory autoclave. Because the costs vary widely based on the size of the instrument, manufacturer, distributor and even the country where it is purchased (I understand that import taxes in Sri Lanka can make the same model more expensive to purchase there than it would be in Canada), we have kept this statement general and high level.

- Expand the limitations section to note the single model tested, potential nutrient degradation with prolonged heating, small production volumes, and electrical compatibility issues in different regions.

• These are also very important points. We have added some language to the manuscript introduction, including reference to previously published paper, which hopefully clarifies why we chose to use this particular model of pressure cooker.

• With respect to nutrient degradation, assessing for the chemical changes that occur during pressure cooking vs. autoclaving was beyond the scope of this study. Rather, we acknowledge that these changes occur in our justification for conducting our experiments as we aimed to determine if the media produced would be fit for purpose.

• The small production volumes and electrical compatibility issues are both things which we mentioned in the text - whether either of these would be a barrier to using this pressure cooker is very context dependent. Wording was added to highlight the step-down voltage transformers do potentially add an additional cost.

Reviewer #3: This article presents a highly practical and beneficial approach, particularly for laboratories that face challenges due to the lack of access to an autoclave. The alternative method proposed could serve as a valuable solution for resource-limited settings. However, further research is recommended, as highlighted by the authors, particularly regarding the use of different brands of pressure cookers in media preparation. Brands that are readily available and suitable for local contexts should be evaluated. In addition, the impact of electrical current capacity on the effectiveness and consistency of results should be examined, as it may influence performance. Lastly, the manuscript formatting should be reviewed to ensure it aligns with the journal’s submission guidelines.

• Thank you for these comments, we completely agree that trialing adding different brands of pressure cookers would be an obvious next step. We have added some additional information in the introduction to explain that this particular model was chosen because a previous study demonstrated that it was able to sterilize the Geobacillus spore biological indicator, while other models were not. In the future, we are interested in evaluating appliances which are locally available in Sri Lanka.

• With respect to the 230V vs. 110V, this should not have had an impact on the performance of the pressure cooker described here. We were able to source a step-down transformer, and so we were able to use 110V on the ground in Sri Lanka.

• We have double checked our formatting and identified some errors in the formatting of our figure legends and references which have now been corrected.

---

## [Decision Letter · Decision Letter 1]

4 Nov 2025

Evaluation of a commercial pressure cooker for the preparation of agar media for a diagnostic microbiology laboratory

PONE-D-25-28065R1

Dear Dr. Rubin,

We’re pleased to inform you that your manuscript has been judged scientifically suitable for publication and will be formally accepted for publication once it meets all outstanding technical requirements.

Kind regards,

Tsegaye Alemayehu, Msc

Academic Editor

PLOS ONE

Additional Editor Comments (optional):

Reviewers' comments:

Reviewer's Responses to Questions

**Comments to the Author**

Reviewer #1: All comments have been addressed

Reviewer #3: All comments have been addressed

2. Is the manuscript technically sound, and do the data support the conclusions?

Reviewer #1: Yes

Reviewer #3: Yes

3. Has the statistical analysis been performed appropriately and rigorously?

Reviewer #1: Yes

Reviewer #3: I Don't Know

4. Have the authors made all data underlying the findings in their manuscript fully available?

Reviewer #1: Yes

Reviewer #3: Yes

5. Is the manuscript presented in an intelligible fashion and written in standard English?

Reviewer #1: Yes

Reviewer #3: Yes

Reviewer #1: Greetings,

Good work.

The revisions have significantly improved the manuscript. It is now scientifically stronger and suitable for publication.

Kind regards,

Reviewer #3: (No Response)

**Do you want your identity to be public for this peer review?** For information about this choice, including consent withdrawal, please see our Privacy Policy

Reviewer #1: No

Reviewer #3: No

---

## [Editor Report · Acceptance letter]

PONE-D-25-28065R1

PLOS ONE

Dear Dr. Rubin,

I'm pleased to inform you that your manuscript has been deemed suitable for publication in PLOS ONE. Congratulations! Your manuscript is now being handed over to our production team.

Kind regards,

on behalf of

Dr. Tsegaye Alemayehu

Academic Editor

PLOS ONE